# ENERGY-INSPIRED SELF-SUPERVISED PRETRAINING FOR VISION MODELS

**Ze Wang**[†]    **Jiang Wang**[‡]    **Zicheng Liu**[‡]    **Qiang Qiu**[†]
[†]Purdue University    [‡]Microsoft Corporation
{zewang, qqiu}@purdue.edu {jiangwang, zliu}@microsoft.com

## ABSTRACT

Motivated by the fact that forward and backward passes of a deep network naturally form symmetric mappings between input and output representations, we introduce a simple yet effective self-supervised vision model pretraining framework inspired by energy-based models (EBMs). In the proposed framework, we model energy estimation and data restoration as the forward and backward passes of a single network without any auxiliary components, e.g., an extra decoder. For the forward pass, we fit a network to an energy function that assigns low energy scores to samples that belong to an unlabeled dataset, and high energy otherwise. For the backward pass, we restore data from corrupted versions iteratively using gradient-based optimization along the direction of energy minimization. In this way, we naturally fold the encoder-decoder architecture widely used in masked image modeling into the forward and backward passes of a single vision model. Thus, our framework now accepts a wide range of pretext tasks with different data corruption methods, and permits models to be pretrained from masked image modeling, patch sorting, and image restoration, including super-resolution, denoising, and colorization. We support our findings with extensive experiments, and show the proposed method delivers comparable and even better performance with remarkably fewer epochs of training compared to the state-of-the-art self-supervised vision model pretraining methods. Our findings shed light on further exploring self-supervised vision model pretraining and pretext tasks beyond masked image modeling.

## 1 INTRODUCTION

The recent rapid development of computation hardware and deep network architectures have paved the way for learning very large deep networks that match and even exceed human intelligence on addressing complex tasks (Brown et al., 2020; He et al., 2017; Silver et al., 2016). However, as annotating data remains costly, leveraging unlabeled data to facilitate the learning of very large models attracts increasing attention. Exploiting context information in the massive unlabeled data in natural language processing (NLP) stimulates Chen et al. (2020a) to use the direct modeling of pixel sequences as the pre-text tasks of vision model pretraining. Recent self-supervised vision model pretraining through masked image modeling (MIM) (He et al., 2022; Wei et al., 2021; Xie et al., 2022) typically adopt an auto-encoder (AE) architecture, where the target vision model to be pretrained serves as an encoder to encode an image with incomplete pixel information to a latent representation. An auxiliary decoder is jointly trained to restore the missing information from the latent representation. Contrastive self-supervised learning methods (Chen et al., 2020b) usually require large training batch sizes to provide sufficient negative samples. Recent Siamese network based self-supervised learning methods (Grill et al., 2020; Chen & He, 2021; Tian et al., 2021; He et al., 2020; Chen et al., 2021) alleviate the huge batch challenge by deploying an momentum copy of the target model to facilitate the training and prevent trivial solutions. VICReg (Bardes et al., 2022) prevents feature collapsing by two explicit regularization terms. Barlow Twins (Zbontar et al., 2021) reduce the need of large batch size or Siamese networks by proposing a new objective based on cross-correlation matrix between features of different image augmentations.

In this paper, we make a further step towards the following question: *Can we train a standard deep network to do both representation encoding and masked prediction simultaneously, so that no auxiliary components, heavy data augmentations, or modifications to the network structure are required?*

Hinted by the fact that the forward and the backward passes of a deep network naturally form symmetric mappings between input and output representations, we extend the recent progress on energy-based models (EBMs) (Xie et al., 2016; Du & Mordatch, 2019; Du et al., 2020b; Zhao et al., 2017) and introduce a model-agnostic self-supervised framework that pre-trains any deep vision models. Given an unlabeled dataset, we train the **forward pass** of the target vision model to perform discriminative recognition. Instead of instance-wise classification as in contrastive self-supervised learning, we train the target vision model to perform binary classification by fitting it to an energy function that assigns low energy values to positive samples from the dataset and high energy values otherwise. And we train the **backward pass** of the target vision model to perform conditional image restoration as in masked image modeling methods, by restoring positive image samples from their corrupted versions through conducting gradient-based updating iteratively along the direction of energy minimization. Such conditional sampling schemes can produce samples with satisfying quality using *as few as one gradient step*, thus prevents the unaffordable cost of applying the standard implicit sampling of EBMs on high-dimensional data. In this way, we naturally fold the encoder-decoder architecture widely used in masked image modeling into the forward and backward passes of a single vision model, so that the structure tailored for discriminative tasks is fully preserved with *no auxiliary components or heavy data augmentation* needed. Therefore the obtained vision model can better preserve the representation discriminability and prevent knowledge loss or redundancy. Moreover, after folding the corrupted data modeling (encoder) and the original data restoration (decoder) into a single network, the proposed framework now accepts a broader range of pretext tasks to be exploited. Specifically, we demonstrate that beyond typical masked image modeling, the proposed framework can be easily extended to learning from patch sorting and learning from image restoration, e.g., super-resolution and image colorization.

We demonstrate the effectiveness of the proposed method with extensive experiments on ImageNet-1K. It is easy to notice that almost every parameter trained from the self-supervised training stage will be effectively used in the downstream fine-tuning. And we show that competitive performance can be achieved even with only 100 epochs of pretraining on a single 8-GPU machine.

## 2    RELATED WORK

**Vision model pretraining.** Pretraining language Transformers with masked language modeling (Kenton & Toutanova, 2019) has stimulated the research of using masked image modeling to pretrain vision models. BEIT (Bao et al., 2021) trains the ViT model to predict the discrete visual tokens given the masked image patches, where the visual tokens are obtained through the latent code of a discrete VAE (Ramesh et al., 2021). iBoT (Zhou et al., 2022) improves the tokenizer with a online version obtained by a teacher network, and learns models through self-distillation. Masked auto-encoder (He et al., 2022) adopts an asymmetric encoder-decoder architecture and shows that scalable vision learners can be obtained simply by reconstructing the missing pixels. (Wei et al., 2021) studies empirically self-supervised training through predicting the features, instead of the raw pixels of the masked images. Different forms of context information for model pretraining are also discussed by learning from predicting the relative position of image patches (Doersch et al., 2015), sorting sequential data (Noroozi & Favaro, 2016), training denoising auto-encoders (Vincent et al., 2008), image colorization (Zhang et al., 2016), and image inpainting (Pathak et al., 2016). Similar to metric learning (Hinton, 2002), contrastive self-supervised learning methods learn visual representations by contrasting positive pairs of images against the negative pairs. (Wu et al., 2018) adopts noise-contrastive estimation to train networks to perform instance-level classification for feature learning. Recent methods construct positive pairs with data augmentation (Chen et al., 2020b), and obtain pretrained models with high discriminability (Caron et al., 2021). To relax the demand of large batch size for providing sufficient negative samples, (He et al., 2020; Chen et al., 2020c) are proposed to exploit supervision of negative pairs from memory queues. And it is shown that self-supervised learning can even be performed without contrastive pairs (Grill et al., 2020; Chen & He, 2021; Tian et al., 2021) by establishing a dual pair of Siamese networks to facilitate the training. (Donahue & Simonyan, 2019) extends unsupervised learning with generative adversarial networks to learning discriminative features.

**Energy-based models.** The proposed framework for vision model pre-training is inspired by the progress of energy-based models (LeCun et al., 2006). As a family of generative models, EBMs are mainly studied to perform probabilistic modeling over data (Ngiam et al., 2011; Qiu et al., 2019; Nijkamp et al., 2020; Du & Mordatch, 2019; Du et al., 2020b; Zhao et al., 2020; Xie et al., 2016; 2017; 2020; 2021; Xiao et al., 2020; Arbel et al., 2021), and conditional sampling (Du et al., 2020a; 2021).

Energy minimization

Energy minimization

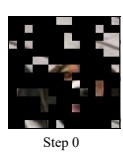 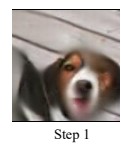 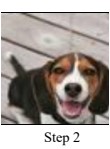

| Step 0 | Step 40 | | Step 0 | Step 1 | Step 2 |

Figure 1: Typical EBM sampling demands long chains even with a mild resolution of $64 \times 64$ (left). Our conditional sampling with short chains obtain satisfactory results with as few as a single gradient step at a standard resolution of $224 \times 224$ (right).

It is shown in (Grathwohl et al., 2020) that a standard discriminative classifier can be interpreted as an EBM for the joint data-label distribution, which can then by exploited to learn from unlabeled data in an semi-supervised manner. Recently, the idea of EBMs is being applied to more applications including reasoning (Du et al., 2022), latent space modeling of generative models (Pang et al., 2020), and anomaly detection (Wang et al., 2022; Dehaene et al., 2020). To the best of our knowledge, we are the first to apply energy-based model training to self-supervised vision model pre-training.

## 3 METHOD

In this section, we introduce in details the proposed framework of energy-inspired self-supervised vision model pretraining. We first briefly review the backgrounds of energy-based models in Section 3.1. We present the general process of the proposed pretraining framework, with a straightforward example based on mask image modeling in Section 3.2. We then present how the proposed framework allows extensions to a wide range of variants adopting different pretext tasks with examples of learning from image restoration (Section 3.3) and learning from sorting (Section 3.4).

### 3.1 BACKGROUNDS

Being mainly generative models, EBMs are usually trained to model a target distribution density function. EBM training is typically achieved by learning an energy function that predicts the unnormalized density, named the energy score, for a given data sample. Specifically, given a data sample $\mathbf{x} \in \mathbb{R}^d$, the energy function $E_\theta(\mathbf{x}) : \mathbb{R}^d \to \mathbb{R}$, with $\theta$ as the learnable parameters, maps the sample to its energy score, which is expected to be low for the in-distribution (positive) samples, and high for the out-of-distribution (negative) samples. The modeled data density $p_\theta(\mathbf{x})$ is expressed as:

$$p_\theta(\mathbf{x}) = \frac{\exp(-E_\theta(\mathbf{x}))}{Z_\theta}, \tag{1}$$

where $Z_\theta = \int_{\mathbf{x}} \exp(-E_\theta(x))$ is the partition function. Approximating a target data distribution $p_{\text{data}}(\mathbf{x})$ equals to minimizing the expected negative log-likelihood function over the data distribution, defined by the maximum likelihood loss function:

$$\mathcal{L}_{\text{ML}} = \mathbb{E}_{\mathbf{x} \sim p_{\text{data}}(\mathbf{x})}[-\log p_\theta(\mathbf{x})] = \mathbb{E}_{\mathbf{x} \sim p_{\text{data}}(\mathbf{x})}[E_\theta(\mathbf{x}) + \log Z_\theta]. \tag{2}$$

As the computation of $\mathcal{L}_{\text{ML}}$ involves the intractable $Z_\theta$, the common practice is to represent the gradient of $\mathcal{L}_{\text{ML}}$ as,

$$\nabla_\theta \mathcal{L}_{\text{ML}} = \mathbb{E}_{\mathbf{x}^+ \sim p_{\text{data}}(\mathbf{x})}[\nabla_\theta E_\theta(\mathbf{x}^+)] - \mathbb{E}_{\mathbf{x}^- \sim p_\theta(\mathbf{x})}[\nabla_\theta E_\theta(\mathbf{x}^-)]. \tag{3}$$

The objective in (3) train the model $\mathbb{E}_\theta$ to effectively distinguish in-domain and out-of-domain samples by decreasing the predicted energy of positive data samples $\mathbf{x}^+$ from the true data distribution and increasing the energy of negative samples $\mathbf{x}^-$ obtained through sampling from the model $p_\theta$.

Sampling from the modeled distribution equals to finding the samples with low energy scores. Parametrizing the energy function as a deep neural network allows for a continuous energy space to be learned from data, where sampling can be accomplished by randomly synthesizing a negative sample of high energy, and moving it in the corresponding energy space along the direction of energy minimization. Inspired by MCMC based sample techniques such Langevin dynamics (Welling & Teh, 2011), common practice (Du & Mordatch, 2019; Du et al., 2020b) resorts to gradient-based

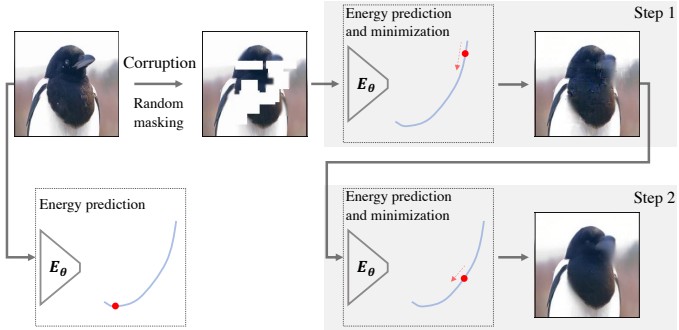

Figure 2: Applying the proposed framework to masked image modeling. The unlabeled image is corrupted with random patches, and the network is trained to recognize the corrupted sample as a negative one with high energy, and recover the original image by updating the image iteratively along the direction of energy minimization.

optimization for implicit sampling. Specifically, by performing $N$ gradient steps, the approximated optimum $\tilde{\mathbf{x}}^N$ can be obtained as

$$\tilde{\mathbf{x}}^n = \tilde{\mathbf{x}}^{n-1} - \alpha \nabla_{\mathbf{x}} E_\theta(\tilde{\mathbf{x}}^{n-1}) + \omega^n, \quad \omega^n \sim \mathcal{N}(0, 2\alpha), \quad n = 1, \dots N, \tag{4}$$

where $\alpha$ is the step size of the gradient-based optimization. In practice, the noise term $\omega^n$ is usually set to a smaller empirical scale as in the official implementation of (Du et al., 2020b) for faster sampling. $\tilde{\mathbf{x}}^0$ is usually obtained by sampling from a predefined prior distribution such as Uniform. For a more comprehensive formulation of implicit generation with energy-based models, please refer to (Xie et al., 2016; Du & Mordatch, 2019; Du et al., 2020b).

### 3.2 PROPOSED FRAMEWORK

We denote the deep vision model to be pretrained as $\psi$. Our energy-inspired model can be constructed by simply appending a linear head $h$ with a single output dimension to the feature extractor, i.e., $E_\theta(\mathbf{x}) = h(\psi(x))$ with $\theta$ collectively denoting the parameters of both $\psi$ and $h$. In a typical setting, the linear head $h$ contains only hundreds of parameters. After the pretraining, the obtained vision model can be directly used as an image recognition model by only replacing the linear head $h$. The full preservation of network architecture with no auxiliary network components, e.g., a decoder, to be removed, better maintains the network discriminability and prevents potential feature redundancy.

As illustrated in Figure 1, even using a low resolution, the typical implicit sampling of EBMs in (4) can take dozens or even hundreds of gradient steps to produce an image sample of satisfying quality (Du & Mordatch, 2019; Zhao et al., 2020). Applying the standard EBM training to self-supervised pretraining introduces unaffordable cost. It is discussed in (Gao et al., 2021) that a reformulation of the training objective based on recovery likelihood can stabilize the training of EBMs. In this paper, inspired by (Wang et al., 2022), we forgo the from-scratch sampling and train the network to perform conditional sampling, so as to restore partially corrupted data with explicit supervision. As visualized in Figure 1, the costly noise-to-image sampling of EBMs is now replaced with conditional sampling, where a chain of sampled data moving towards the low-energy region are obtained for each corrupted sample rapidly. In our case of self-supervised learning, doing so has two major advantages: Firstly, as being further discussed in Section 4.3, the proposed framework now allows the restoration of each sample to be completed with as few as two gradient optimization steps, and permits desirable speed for self-supervised training on large scale datasets. Moreover, such conditional sampling allows us to replace the contrastive divergence (3) designed for unconditional sampling by explicit supervision with pixel values as we will discuss later, and such strong supervision alleviates the unstable EBMs training according to our observations. The proposed framework imposes little restrictions to the image sample corruption methods deployed and permits a wide range of pretext tasks to be exploited. For the sake of discussion, we present in details one straightforward variant with masked image modeling to walk through the training process, and illustrate other possible variants in later sections.

**Masked image modeling.** As visualized in Figure 2, given a batch of image samples $\{\mathbf{x}_i\}_{i=1,\dots,K}$, we first corrupt each image using a predefined function $\downarrow(\cdot)$. In this example, $\downarrow(\cdot)$ denotes random image masking. After image masking, $\downarrow(\mathbf{x}_i)$ can be seen as a sample that is out of the target data distribution $p_{\text{data}}$ with the remaining pixels inferring the original contents of the image. With the target modeling a continuous energy function, we can perform online evaluation to the estimated energy function by examining how well moving the masked image in the modeled energy space

along the energy minimization direction can restore the original data $\mathbf{x}_i$. Specifically, we resort to the gradient based optimization (4) and perform $N$-step image restoration with $\tilde{\mathbf{x}}_i^0 = \downarrow (\mathbf{x}_i)$. The loss of the restoration steps can then be expressed as:

$$\mathcal{L} = \frac{1}{KN} \sum_{i=0}^{K} \sum_{j=0}^{N} \text{MSE}(\tilde{\mathbf{x}}_i^j, \mathbf{x}_i), \quad \text{where} \quad \tilde{\mathbf{x}}_i^j = \tilde{\mathbf{x}}_i^{j-1} - \alpha \nabla_{\mathbf{x}} E_\theta(\text{SG}(\tilde{\mathbf{x}}_i^{j-1})), \qquad (5)$$

with SG denoting the *stop gradient* operation that blocks the gradient propagation across steps. We empirically observe that adding stop gradient operations between consecutive steps helps accelerate the training speed and convergence. $\mathcal{L}$ here encourages original images to be restored from the negative images (corrupted versions and the sampled versions along the sampling chains of (5)) by gradient based updating along the direction of energy minimization, which equally encourages higher energy values for negative images, and can functionally replace the second term in (3). The supervision in (5) is similar to the ones used in score matching and diffusion models (Vincent, 2011; Ho et al., 2020). One major different is that all the inputs for the intermediate steps in our method are obtained by the previous step of restoration, instead of generated using the original images based on some noise schedulers.

Notably, as discussed in (Du & Mordatch, 2019), standard EBM training with (3) using arbitrary energy model can cause sharp changes in gradients, and the stable training requires heavy tuning to the hyperparameters and techniques like spectral normalization to constrain the Lipschitz constant of the network. While in our framework, unstable training caused by sharp gradients is naturally prevented by the explicit supervision in (5), as faithfully restoring the original data requires the gradient in (5) to be bounded within a certain range. We summarize the overall training steps of the proposed framework in Algorithm 1. We further provide PyTorch-style pseudo code in Appendix Section A.3 to facilitate reproducing our results.

---

**Algorithm 1** Energy-based self-supervised vision model pretraining.

---

1: **Given**: A target network $\psi$ to be pretrained, a large-scale unlabeled dataset $\{\mathbf{x}_i\}$, and an image sample corruption function $\downarrow (\cdot)$.
2: **Given**: Step size $\alpha$ and number of steps $N$ for the gradient update of corrupted samples.
3: Initialize the target network $\psi$ and the linear head $h$.
4: **repeat**
5:     Sample a batch of images from the unlabeled dataset.
6:     Corrupt each sample and initialize the conditional sampling chains as $\tilde{\mathbf{x}}_i^0 = \downarrow (\mathbf{x}_i)$.
7:     **for** Step $n = 1 : N$ **do**
8:         Stop gradient $\tilde{\mathbf{x}}_i^{n-1} = \text{SG}(\tilde{\mathbf{x}}_i^{n-1})$.
9:         Perform gradient update to the corrupted samples as in (5).
10:     **end for**
11:     Compute the restoration error of each step using (5), and update $\psi$ and $h$ with gradient optimization.
12: **until** Converge
13: **Return** $\psi$.

---

### 3.3 Beyond Mask Image Modeling

Recent self-supervised vision model pretraining methods (Xie et al., 2022; He et al., 2022; Wei et al., 2021) invariably adopt masked image modeling as the pretext task. We argue that *the encoder-decoder architectures used in these methods prevent them from being easily extended to other pretext tasks*. In the auto-encoder based methods, the vision model to be pretrained serves as the encoder, and is only exposed with the corrupted images during pretraining. Therefore, it is important to present part of the original image patches to the encoder, so that the encoder can learn from those intact patches network weights that transfer well in downstream finetuning. While in the proposed pretraining framework, both corrupted samples and original samples are exposed to the target vision model, in the forms of input and supervision, respectively. Specifically, by simply replacing the corruption function $\downarrow (\cdot)$, we can establish a wide range of pretext variants that learn vision models from, such as *patch sorting, super-resolution, denoising, and image colorization*. Further details and results will be discussed in Section 4.1. With certain degrees of global image corruption, networks can be trained to infer possible content given the incomplete pixel information, and restore the missing information, such as detailed textures or color, by the patterns learned from the true data and stored in the network weights. With the restriction to the corruption methods being lifted, our framework stimulates further discussions on the pretext tasks of vision model pretraining. Patch sorting is discussed next as an example of new pretext tasks.

### 3.4 LEARNING FROM SORTING

Sorting the patches of an image according to the spatial position requires inferring the global content by integrating the local information contained in each patch, and sorting the order of the patches accordingly. Such process involves both local feature extraction and global semantic inference, therefore can be an useful pretext task of self-supervised training. However, restoring the patch orders in the image pixel space can be extremely challenging to learn. MP3 (Zhai et al., 2022) extends mask image modeling to position predictions by dropping tokens in the values of the self-attention layers and predicting the corresponding position using the extracted features. Thanks to the absolute position embedding widely adopted in ViTs, we present an interesting variant of learning from sorting that does not modify any intermeidate layers of ViTs.. Specifically, the feature extraction of a ViT $\psi$ can be expressed as:

$$\psi(\mathbf{x}_i) = \phi_{\mathrm{T}}(\mathbf{z}_{\mathrm{class}}, \{\phi_{\mathrm{P}}(\mathbf{x}_i[p,q]) + \mathrm{PE}[p,q]\}_{p=1,q=1}^{P,Q}) \tag{6}$$

where $\phi_{\mathrm{T}}$ and $\phi_{\mathrm{P}}$ denote the stacked transformer layers and the patch embedding layer, respectively. We use $p$ and $q$ to index the image patch and $\mathrm{PE}[p,q]$ is the corresponding position embedding. Adopting the simple non-learnable sin-cos function as the position embedding, we can shuffle the image patches by simply shuffling the position embedding, and train the target network to sort the patches by performing gradient-based optimization to the shuffled position embedding along the direction of energy minimization. Specifically, based on (6) and omitting the indexes, we define the new energy function parametrized by the target vision model as $E_\theta(\mathbf{x}_i, \mathrm{PE})$, and train the network using the following loss

$$\mathcal{L}_{\mathrm{sort}} = \frac{1}{KN} \sum_{i=0}^{K} \sum_{j=0}^{N} \mathrm{MSE}(\tilde{\mathrm{PE}}_i^j, \mathrm{PE}), \quad \text{where} \quad \tilde{\mathrm{PE}}_i^j = \tilde{\mathrm{PE}}_i^{j-1} - \alpha \nabla_{\mathrm{PE}} E_\theta(\mathbf{x}_i, \mathrm{SG}(\tilde{\mathrm{PE}}^{j-1})). \tag{7}$$

Learning from sorting corrupts only the position embedding, and allows the original image signal to be fully exposed to the network. And the network is encouraged to infer the global structure of an image from the features of patches and sort the patches to form a semantically meaningful pattern.

**Random edge masking in the patch embedding layer.** Note that for most natural images, two neighboring patches may share nearly identical pixels at the edge rows or columns. While such information is hard to be exploited by human for patch sorting, is can be easy for a network to learn such trivial sorting solution, and perform nearly perfect patch sorting without resorting to actual semantics. Such trivial solution can be easily avoided by random masking to the weights in the patch embedding layer and preventing the patch embedding layer from learning only the edge pattern of image patches. We discuss the detailed implementation of the edge masking in Appendix Section A.2. Such trivial solution can also be resolved by randomly masking tokens in vision transformers as in (Zhai et al., 2022).

## 4 EXPERIMENTS

With the standard ImageNet-1K dataset, we show that the proposed EBM pretraining framework can help a deep vision model to achieve competitive performance with as few as 200 epochs of training. We use ViT to conduct most of the experiments. And we further show in Appendix Section B.3 that as a model-agnostic framework, the proposed method can be seamlessly extended to other architectures.

**Training.** We use AdamW (Loshchilov & Hutter, 2019) as the optimizer for both self-supervised training and tuning. For all the self-supervised pretraining experiments, we adopt only random cropping and random horizontal flipping as the data augmentation. We present comprehensive training details in Appendix Section A.1 Table A. Most of the experimental settings follow (He et al., 2022). Unlike recent methods (Zhou et al., 2022; He et al., 2022), we *do not* perform exhaustive searches for the optimal hyperparameters such as learning rates. Training energy functions introduces a new hyperparameter $\alpha$, which is the step size of the gradient optimization to the corrupted data. Thanks to the explicit supervision available in the proposed framework, we can set $\alpha$ to be learnable, and jointly train it with the network without the concern of training stability as in standard EBM training. If not otherwise specified, we adopt $N = 2$, i.e., two steps of gradient-based energy minimization in the pretraining stage for the best performance-efficiency trade-off.

### 4.1 SELF COMPARISONS

As discussed in Section 3, the proposed framework accepts a wide range of variants with different pretext tasks. To illustrate the flexibility, we present results with different variants including learning

Table 1: Masked image modeling with different patterns and ratios of image masking. The result of MAE (He et al., 2022) with 400 epochs is based on our reimplementation. The results of our methods are obtained by 100 epochs of pretraining. All results are obtained with 100 epochs of finetuning. Baseline results are in gray. *From scratch* indicates the purely supervised baseline.

| Masking strategies | | Accuracy | | | |
|---|---|---|---|---|---|
| From scratch | | 76.6 | | | |
| Random large | | 79.7 | | | |
| Random small | | 79.3 | | | |
| % of masking | 10% | 30% | 50% | 70% | 90% |
| MAE | - | - | - | 78.3 | - |
| Gridded (16) | 76.7 | 78.3 | 78.7 | 79.0 | 78.8 |
| Gridded (24) | 76.8 | 78.2 | 78.7 | 79.2 | 78.8 |
| Gridded (32) | 77.1 | 78.4 | 78.6 | 79.0 | 78.7 |

Table 2: Results obtained by different pretext tasks of learning from image restoration and patch sorting. Baseline results are in gray.

| Methods | Accuracy |
|---|---|
| From scratch | 76.6 |
| AE + SR 16× | 77.1 |
| AE + denoising | 76.8 |
| Denoising | 79.2 |
| SR 8× | 77.1 |
| SR 14× | 78.2 |
| SR 16× | 79.6 |
| SR 24× | 78.4 |
| SR 32× | 76.3 |
| Colorization | 79.5 |
| AE + sorting | 77.2 |
| Patch soring | 79.5 |

from masked image modeling, image restoration, and sorting. All results in this section are obtained by pretraining and finetuning a ViT-S for 100 epochs on the ImageNet-1K (Deng et al., 2009) dataset.

**Learning from masked image modeling.** A straightforward way of implementing the proposed framework is to train the network to perform masked image modeling given incomplete pixel information. We present results obtained with different masking strategies and ratios of masking in Table 1. As visualized in Figure 3, in the experiments with gridded mask, we evenly divide an image into squared patches with the same size, and randomly mask out a portion of the patches. Note that in the *Gridded (16)* experiments, the patch partition in the image masking matches exactly with the patch partition in the ViT networks, therefore it is a fair comparison against MAE (He et al., 2022). For the random masking experiments, we randomly place blank patches with the size and aspect ratio sampled from a particular range to each image. In the *Random small* experiments, we randomly place 75 blank patches with normalized sizes sampled from a Uniform distribution of $\mathcal{U}(0.01, 0.025)$. In the *Random large* experiments, we randomly place 25 blank patches with normalized sizes sampled from $\mathcal{U}(0.02, 0.05)$. For both experiments, the aspect ratio of each patch is sampled from $\mathcal{U}(0.5, 2.0)$.

**Learning from image restoration.** As discussed in Section 3.3, our framework enjoys higher flexibility as the pretrained vision model is exposed with both true samples and artificial negative ones, thus even when the input images are corrupted globally, our framework can still learn good models. To show this, we present in Table 2 results obtained with learning from image restoration. Specifically, we train the network to learn from *image super-resolution, denoising, and image colorization*, where every pixel is corrupted with a predefined function. Table 2, *SR* denotes super-resolution. *AE + SR 16* denotes a baseline experiment with a auto-encoder architecture as in (He et al., 2022). In the $s$-time super-resolution (denoted as *SR $s\times$*), the image are first downsampled using bicubic interpolation for $s$ times, and resized back to the original size using nearest-neighbor interpolation. In the denoising experiments, we take a noise scheme inspired by diffusion models (Song et al., 2021a; Ho et al., 2020) with $\downarrow(\mathbf{x}) = \sqrt{\gamma}\mathbf{x} + \sqrt{1-\gamma}\epsilon$, with $\epsilon \sim \mathcal{N}(0, I)$ and $\gamma$ uniformly sampled as $\gamma \sim \mathcal{U}(0, 1)$.

As shown in the quantitative results in Table 2 and visualizations in Figure 3, with proper degrees of corruption, restoring the original images may require the network to infer the general content given the corrupted pixels, and recover the details using the knowledge learned from the true samples and stored in the network weights. For example, in the image colorization experiments, the pretrained vision model learns the common colors of different objects from the massive unlabeled data in a self-supervised way. As visualized in Figure 3, the vision model learns common knowledge such as stop signs are usually red, and the background of a horse is usually green while manatees are marine mammals therefore the background is usually blue. Summarizing such knowledge requires the vision models to learn identifying objects first, therefore transferable network weights and feature representations can be obtained from pretraining. And as shown in the 'Denoising' row of Figure 3, with strong noise injected to the input, the model is able to recover objects that are almost invisible. This finding potentially connects our pretraining method with genitive models (Song et al., 2021a; Ho

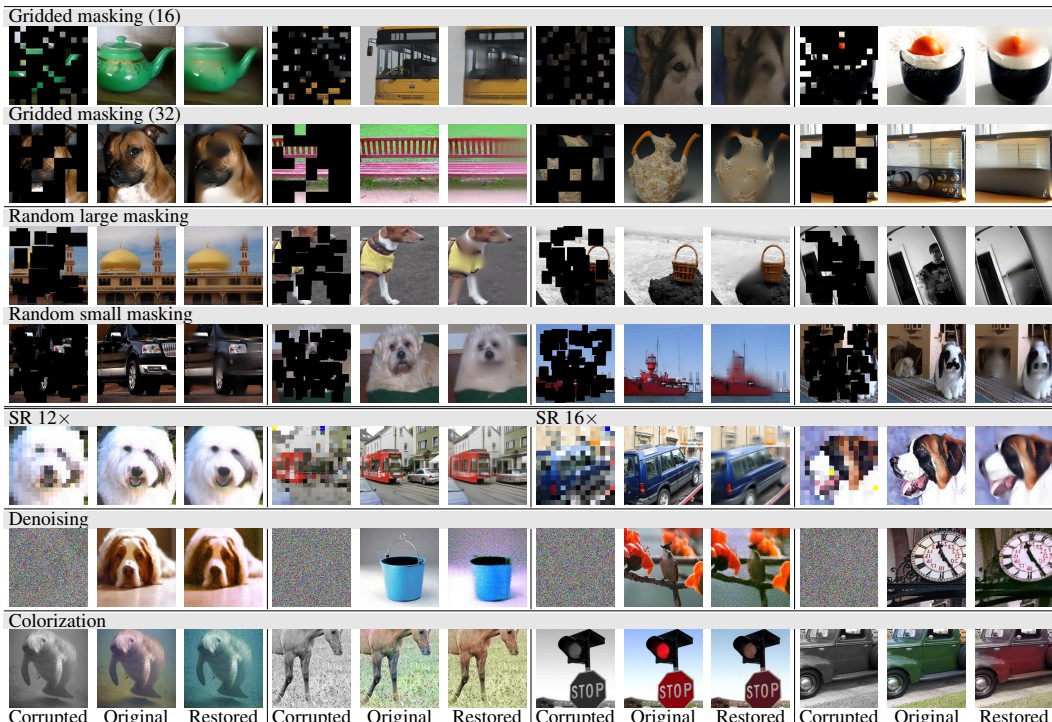

Gridded masking (16)

Gridded masking (32)

Random large masking

Random small masking

SR 12×              SR 16×

Denoising

Colorization

Corrupted  Original  Restored  Corrupted  Original  Restored  Corrupted  Original  Restored  Corrupted  Original  Restored

Figure 3: Conditional sampling with masked image modeling with different masking strategies and learning from image restoration. The proposed framework accepts a broader range of pretext tasks.

Table 3: Quantitative comparisons against the recent self-supervised model pretraining methods. ∗ denotes results produced by our re-implementation. PT and FT denote pretraining and finetuning, respectively. All ImageNet results are evaluated on the validation set with a single center crop of 224×224 for each image. † denotes the training involves external dataset other than ImageNet-1K. For our results, we set $e = 50$ for ViT-L, $e = 100$ for ViT-B, and $e = 200$ for ViT-S. We use $N = 2$ for 200-epoch experiments and $N = 1$ otherwise.

Table 4: ViT-S training efficiency with number of epochs. Results of MAE (He et al., 2022) are obtained with our re-implementation.

| Methods | Epochs | Accuracy |
|---|---|---|
| MAE | 400 + 100 | 78.3 |
| MAE | 400 + 200 | 80.2 |
| Ours | 100 + 100 | 79.7 |
| Ours | 100 + 200 | 81.0 |

Table 5: Efficiency comparisons with GPU-hours. † denotes numbers obtained by (Zhou et al., 2022).

| Methods | GPUs × H | Acc. |
|---|---|---|
| MoCo-V3 | 128 × 24 | 83.2 |
| BEiT† | 16 × 90 | 81.4 |
| DINO† | 16 × 112 | 81.6 |
| iBOT | 16 × 193 | 82.0 |
| MAE (400+100) | 8 × 72 | 83.1∗ |
| MAE (1600+100) | 8 × 288 | 83.6 |
| Ours | 8 × 232 | 83.4 |

| Methods | (PT + FT) | ViT-S | ViT-B | ViT-L |
|---|---|---|---|---|
| From scratch | 300 | 79.6∗ | 82.3 | 82.6 |
| DINO | - | - | 82.8 | - |
| MoCo-V3 | 300+150 | - | 83.2 | 84.1 |
| BEiT† | 800+100 | - | 83.2 | 85.2 |
| MaskFeat | 300+100 | - | 83.6 | 85.7 |
| iBOT | 600 + 200 | 81.4 | - | - |
| iBOT | 1600 + 100 | - | 83.8 | - |
| MP3 | 100 (150) + 100 | - | 83.0 | 83.6 |
| MSN | 600 + 100 | - | 83.4 | - |
| MAE | 400 + 100 | 78.3∗ | 83.1∗ | - |
| MAE | 1600 + e | - | 83.6 | 85.9 |
| **Ours Sorting** | 800 + 100 | - | 83.2 | - |
| **Ours Mixed** | 200 + e | 81.2 | 83.1 | - |
| **Ours Mixed** | 800 + e | 81.8 | 83.4 | 85.4 |

et al., 2020; Song & Ermon, 2019; Song et al., 2021b). We will further investigate the connections in future efforts. In our method, corrupted images and original image are exposed to the input layers of the vision model as input and supervision, respectively. Therefore, compared to the auto-encoder based baseline, which only receives corrupted images as input, the proposed framework demonstrates clearly better performance after finetuning.

**Learning from sorting.** To prevent the trivial solution of sorting as discussed in Section 3.4, we adopt regularization schemes that prevents the network from simply sorting the patches based on the edge pixels only. Details of the regularizations can be found in Appendix Section A.2. For a fair comparison, we conduct a baseline experiment of adopting an auto-encoder network to directly predict the position of each patch. We follow the details presented in MAE (He et al., 2022) and implement an asymmetric auto-encoder structure with a lightweight decoder. Note that in the baseline implementation, there is no position embedding used in the encoder ViT, and we add back trainable position embedding initialized with full zeros in the finetuning stage for fair comparisons. The quantitative comparisons are in the bottom rows of Table 2. All numbers are obtained with the same settings. We apply the same regularization schemes to the baseline method to prevent trivial solution. The proposed method learns better features. And the discrepancy between pretraining and finetuning caused by the position embedding in the encoder of the AE baseline may be an important reason of its worse performance.

## 4.2 Quantitative Comparisons Against Recent Methods

In this section, we present quantitative comparisons against the recent self-supervised model pretraining methods. We train our method using a mixture of pretext tasks that are uniformly sampled from image masking, super-resolution, denoising, and colorization. In Table 3, we compare our method against DINO (Caron et al., 2021), MoCo-V3 (Chen et al., 2021), MaskFeat (Wei et al., 2021), BEiT (Bao et al., 2021), iBOT (Zhou et al., 2022), MP3 (Zhai et al., 2022), Masked Siamese Networks (MSN) (Assran et al., 2022), and MAE (He et al., 2022). We train our model with a mixture of all the pretext tasks discussed above. We do this by randomly sample a correction method for each image in a batch. With only 200 epochs of pretraining, the proposed framework can achieve comparable or even better performance with the state-of-the-art self-supervised pretraining methods, some of which adopt much more epochs and leverage external data for training.

## 4.3 Efficiency Discussions

In this section, we present discussions on the training efficiency of the proposed method. As discussed in Section 3.2, in the proposed framework, the network learns to model the density with samples from real data distribution and sampled negative. Therefore, compared to other masked image modeling based pretraining methods that learn from a relatively small part of the images in each iteration, the proposed framework delivers comparable performance with fewer epochs of training. Note that when using each training iteration in our approach involves $N$ forward passes and $N + 1$ backward passes with an additional backward pass for the gradient of parameters. We present efficiency comparisons with number of epochs in Table 4. The proposed method can achieve better results with fewer epochs of training compared to MAE (He et al., 2022).We further present comparisons on training efficiency with GPU-hours in Table 5. The proposed method demonstrates a good performance-efficiency trade-off compared to the state-of-the-art methods. We present in Appendix Figure A performance obtained with different $N$ steps of updating.

## 5 Conclusion

We presented energy-inspired self-supervised pretraining for vision models. We accelerated EBM training and trained the vision model to perform conditional sampling initialized from corrupted samples by moving them along the direction of energy minimization. The bi-directional mappings between images and latent representations are modeled naturally by the forward and backward passes of a network, which fully preserve the discriminative structure of the target vision model and avoid auxiliary network components and sophisticated data augmentation to facilitate pretraining. We presented extensive experiments with different pretext tasks, including learning from masked image modeling, learning from image restoration, and learning from sorting. We hope our findings can shed light on further exploring the pretext tasks of self-supervised vision model pretraining.

**Limitation.** While strong finetuning results are observed, our method does not directly provide features that are strongly linearly-separable, which is reflected by lower linear probing accuracy compared to contrastive learning based pretraining methods. This phenomena is also observed and discussed in (He et al., 2022), and may be attributed to the fact that both (He et al., 2022) and our method do not explicitly encourage linear separation of features in the pretraining stage as the contrastive learning based method do. And linear probing cannot faithfully validate strong but non-linear features. We present quantitative comparisons in Appendix Section B.4, and will keep improving the linear separation as a direction of future effort.

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

## A  IMPLEMENTATION DETAILS

### A.1  DETAILS ON TRAINING

We present the training details for both self-supervised training and finetuning in Table A. All experiments are implemented using PyTorch (Paszke et al., 2019). We use the default API for automatic mixed-precision training. The step size $\alpha$ can either be initialized as $0.1$ and trained along with all the parameters in the vision model in an end-to-end fashion, or predicted by a linear head given the extracted features. In practice, we impose a positive constraint to the value of $\alpha$ during training. Following standard practice, we use an image resolution of $224 \times 224$ in all experiments.

| Configurations | Pretraining | Finetuning |
|---|---|---|
| optimizer | AdamW | AdamW |
| base learning rate | 1e-4 | 1e-3 |
| learning rate schedular | Cosine decay | Cosine decay |
| weight decay | 0.05 | 0.05 |
| momentum of AdamW | $\beta_1 = 0.9, \beta_2 = 0.95$ | $\beta_1 = 0.9, \beta_2 = 0.999$ |
| layer-wise lr decay (Clark et al., 2020) | - | 0.75 |
| batch size | 256 | 1024 |
| drop path (Huang et al., 2016) | - | 0.1 |
| augmentation | RandomResizedCrop | RandAug (9, 0.5) (Cubuk et al., 2020) |
| label smoothing (Szegedy et al., 2016) | - | 0.1 |
| mixup (Zhang et al., 2018) | - | 0.8 |
| cutmix (Yun et al., 2019) | - | 1.0 |
| Mix-precision training | ✓ | ✓ |

Table A: Training details for both self-supervised pretraining and finetuning.

### A.2  LEARNING FROM SORTING

In the experiments of learning from sorting, we adopt the 2D sin-cos position embedding following the implementation of MoCo-V3 (Chen et al., 2021). The embedding for each position remains fixed during the self-supervised pretraining stage. In the finetuning stage, we initialize the position embedding using the same 2D sin-cos function, and allow the embedding to be trainable along with all the parameters in ViT.

**Regularization.**  To prevent the trivial solution of sorting the patches based on only the edge pixels of image patches as discussed in Section 3.4, we adopt two regularization methods in the pretraining stage. Firstly, we adopt the random edge masking as presented Section 3.4. Specifically, for each image patch, we randomly set the values of the $k$ out-most pixels to all zeros. We observe that satisfactory performance can be achieved by simply setting the probabilities of $k = 1$ and $k = 2$ to 50% and 50%, respectively. To further improve the robustness and training speed, in each iteration, we randomly dropout 50% of the image patches before the patch embedding enters the Transformer layers. This simple *patch drop-out* scheme significantly reduces chance that neighboring patches entering the Transformer layers simultaneously, and prevents the network from learning to sort the patches simply based on edge pixels of patches.

### A.3  PSEUDO CODE IN PYTORCH STYLE

```
model = VisionModel()
# initialize deep vision model with any architectures
head = Linear(in_channels=model.dim, out_channels=1, bias=False)
# initialize a simple linear head for energy score prediction

criterion = SmoothL1Loss(beta=1.0)
# define loss function for image reconstruction

optimizer = AdamW(model.parameters() + head.parameters())
# initialize parameter optimizer
```

```
12
13  # training loop
14  for images in image_loader:
15      # images with shape [n, c, h, w]
16      corrupted_images = corruption_method(images)
17
18      loss = 0
19
20      for _ in num_steps:
21          corrupted_images = corrupted_images.detach()
22          # stop gradients between inner-loop steps.
23          energy_score = head(model(corrupted_images))
24          # energy score with shape [n, 1]
25
26          im_grad = autograd(energy_score.sum(), corrupted_images)
27          # compute the gradient of input pixels along the direction
28          # of energy maximization
29          corrupted_images = corrupted_images - alpha * im_grad
30          # gradient descent along the direction of energy minimization
31
32          loss += criterion(corrupted_images, images)
33
34      optimizer.zero_grad()
35      loss.backward()
36      optimizer.step()
```

Listing 1: PyTorch-style pseudo code of the proposed pretraining framework.

## B    ADDITIONAL ANALYSIS

### B.1    PERFORMANCE WITH DIFFERENT STEPS OF GRADIENT UPDATE

We present performance obtained with different $N$ steps of gradient update to the corrected samples. We use $N = 2$ for the best performance-efficiency trade-off and the proposed framework can perform fairly well with as few as a single step of gradient update to each corrupted sample.

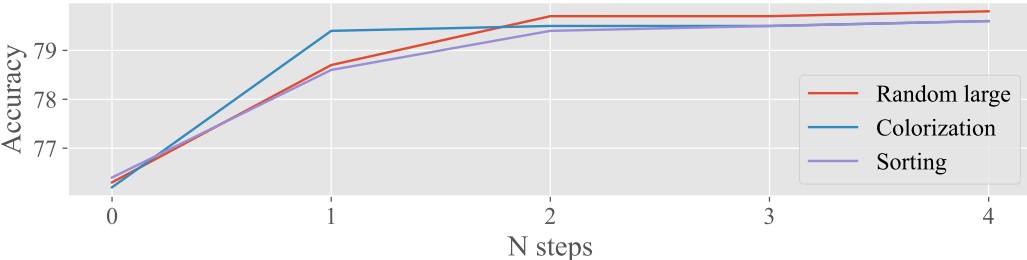

Figure A: Performance with different $N$. $N = 0$ corresponds to using corrupted images as negative.

### B.2    LOSS CURVE

We plot the training curve for the mixed experiment with ViT-B in Figure B. The potentially inaccurate energy estimation at early training stage does not hurt the overall training.

### B.3    OTHER NETWORK ARCHITECTURES AND DOWNSTREAM TRANSFER

Different from models like MAE (He et al., 2022) and SimMIM (Xie et al., 2022) that are specifically tailored for particular network architectures, our framework can be seamlessly applied to any deep vision models without any customization or auxiliary network components beside the simple linear head $h$. To show this, we present results with convolution-based ResNet (He et al., 2016), ConvNeXts (Liu et al., 2022) and Swin-Transformer (Liu et al., 2021) in Table B. We replace the batch normalization layers with group normalization layers in ResNet to ensure the training stability. And to validate

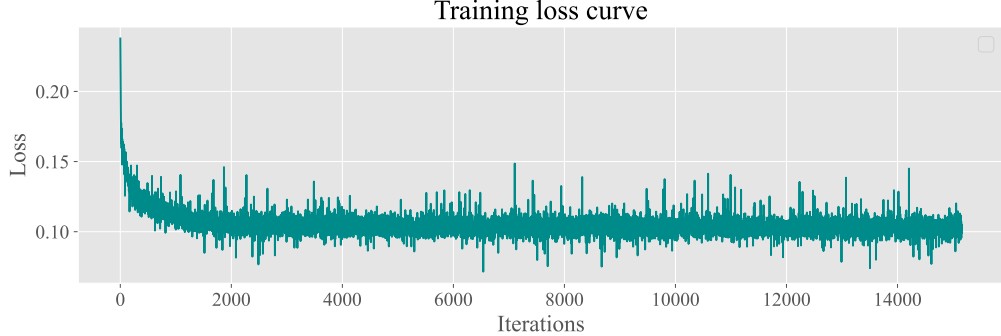

Figure B: Training loss curve.

Table B: The proposed framework can be seamlessly applied to any deep vision models. FS, PT, and FT denote from-scratch training, pretraining, and finetuning, respectively.

| Networks | FS 300E | PT 200E + FT 100E |
|---|---|---|
| ConvNeXt-T | 82.1 | 82.7 |
| Swin-T | 81.3 | 82.2 |
| ResNet-50 | 76.5 | 77.2 |

Table C: mIoU results with ADE20K semantic segmentation finetuning.

| method | data | ViT-B |
|---|---|---|
| Supervised | ImageNet | 47.4 |
| MoCo-v3 | ImageNet | 47.3 |
| BEiT | ImageNet+DALL-E | 47.1 |
| MAE | ImageNet | 48.1 |
| Ours | ImageNet | 47.5 |

the effectiveness to the downstream transfer, we finetune the pretrained network on the ADE20K (Zhou et al., 2017) semantic segmentation dataset, and present the results with mean interaction over union (mIoU) in Table C.The proposed framework generalizes well across architectures and downsteam tasks.

## B.4 ADDITIONAL EVALUATIONS

We report in Table D additional evaluations on ImageNet-1K. All numbers are obtained with the ViT-B network. And we present experiments with a non-linear MLP head (three layers with 768 channels and ReLU activation) with the pretrained feature extractor frozen. For both MAE He et al. (2022) and our method, we consistently observe that a three-layer MLP head cannot significantly improve the performance compared to linear probing.

Following the low-data regime discussions in (Chen et al., 2020b), we further present in Table D results with semi-supervised finetuning on ImageNet. Specifically, we finetune the entire networks with 1% 10% ImageNet training samples and report the top-1 accuracy on the official validation set.

To further evaluate how well the learned features transfer to downstream tasks, following the discussions in (Chen et al., 2020b), we present linear probing results with additional fine-grained natural image datasets in Table D.

## B.5 ENERGY SCORE

In Figure C, we show the histograms of the scores estimated by a trained model. *Step 0* in Figure C corresponds to the scores of the manually corrupted images, and *Step 1* corresponds to the scores of the images obtained by one-step recovery by our model given the manually corrupted images. The trained model assigns lowest scores to the real images. We further present the the energy score histograms of ImageNet and the validation set of Stanford Cars (Over 8K images) in Figure D. The energy estimation generalizes well to unseen images as the network assigns the same low scores to the images of the additional natural images.

Table D: Additional empirical evaluations. We report the top-1 accuracy for all experiments.

| Models | ViT-B + MAE | ViT-B + Ours |
|---|---|---|
| Linear probing | 67.8 | 66.5 |
| MLP head | 68.2 | 67.2 |
| 1% semi-supervised | 48.48 | 48.67 |
| 10% semi-supervised | 72.73 | 72.58 |
| Cars | 52.13 | 52.58 |
| Aircraft | 58.43 | 58.67 |
| Pets | 85.64 | 85.08 |
| Flowers | 93.38 | 94.02 |

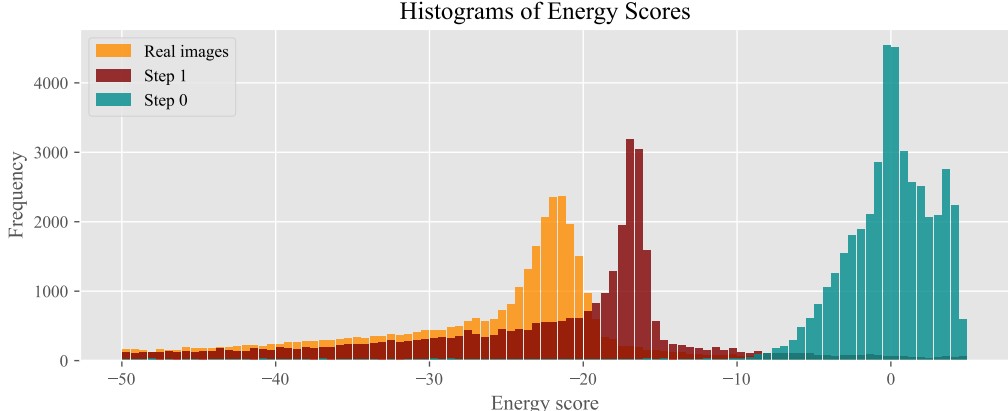

Figure C: Histograms of energy scores. All scores are obtained on the ImageNet validation set.

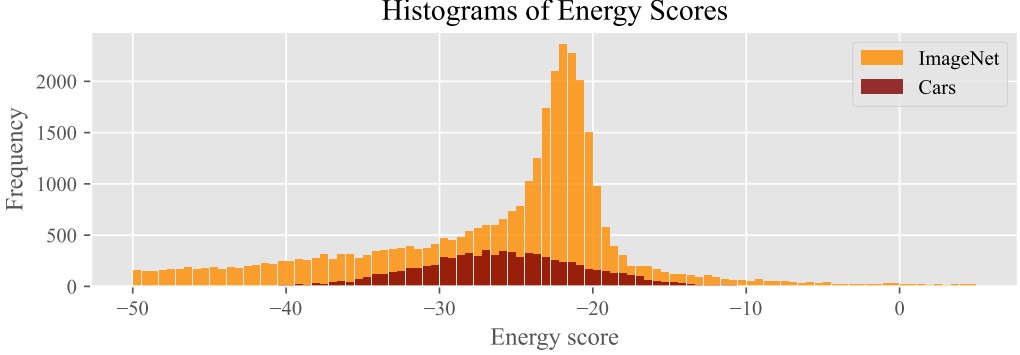

Figure D: Energy score histograms of natural images.

