# OpenReview forum: "Energy-Inspired Self-Supervised Pretraining for Vision Models"
_ICLR.cc/2023/Conference — ICLR 2023 notable top 25%_

### Official Review · Reviewer_qFyR · 2022-10-24

**Confidence:** 4
**Correctness:** 3
**Technical Novelty And Significance:** 3
**Empirical Novelty And Significance:** 3
**Recommendation:** 6

**Clarity, Quality, Novelty And Reproducibility:**

Overall the paper is well written and easy to follow. To my knowledge, the approach is new.
Regarding the model formulation, it would be informative to work out the precise connection of the training objective (5) to the EBM objective (1) if there is one.

In terms of related work, it would probably be worth mentioning iGPT [b] and BigBiGAN [c] as two recent generative modeling based representation learning approaches.

Some details on the experiments are missing, for example I could not find any mention of the image resolution used to train the method. The paper mentions a patch size of 24, which does not divide 224, which is a commonly used training resolution.

Typos:
- P5: “Note that…” instead of “Noted that…”; “...actual semantics.” instead of “...actual semantic.”
- p9: “...images. Different…” instead of “...images.Different…”

[b] Chen, Mark, et al. "Generative pretraining from pixels." International conference on machine learning. PMLR, 2020.

[c] Donahue, Jeff, and Karen Simonyan. "Large scale adversarial representation learning." Advances in neural information processing systems 32 (2019).


**Strength And Weaknesses:**

*Strengths:*
- The approach seems to be novel and takes the direction direction of using generative models for representation learning, which was less successful than e.g. contrastive methods in the recent past
- The method can be combined with several corruptions, which might lead to different representations which are amenable to different types of downstream tasks.
- The method is similar to MAE in that it produces a dense output in image space, but does not require a decoder which is only used in pretraining and does not contribute to the representation.

*Weaknesses:*
- The paper only reports fine-tuning results on the full ImageNet data set (and ADE20k), while acknowledging that training a linear head on top of the frozen representation leads to weaker performance than contrastive methods, similar to MAE. To strengthen the paper, the authors could provide linear evaluation results or results based on an MLP head for reference and comparison with other methods, or - more sensibly - transfer results to different data sets potentially in a low data regime (see, e.g. Chen et al. 2020 for inspiration).
- The authors highlight several times that their method needs fewer steps to reach a given quality than prior work, but they do not discuss that their method requires two forward passes, and three backward passes (two gradient computations w.r.t input, and one w.r.t. the parameters for the update step). I think this should be discussed explicitly (I do acknowledge the walltime evaluation in Table 5).
- Sorting as a self-supervised task has been explored recently in [a]. I feel the paper should discuss [a] and compare their results to [a].

[a] Zhai, Shuangfei, et al. "Position Prediction as an Effective Pretraining Strategy." International Conference on Machine Learning. PMLR, 2022.


**Summary Of The Paper:**

The paper presents a novel representation learning method inspired by energy-based generative models, which can be combined with different restoration tasks as self-supervised surrogate tasks. Specifically, a neural network-based energy function is learned which assigns a high energy to corrupted images, and a low energy to clean images. The method is evaluated for different corruptions and performs competitively with MAE when fine-tuning on the full ImageNet data set, without using a decoder for pretraining.

**Summary Of The Review:**

The proposed method is interesting and novel, and turns many common restoration tasks into self-supervised learning objectives. The method shows promising results in the presented evaluations.

The paper could benefit from additional evaluations, such as transfer beyond the training data distribution, possibly in a low data regime. Furthermore, there are several aspects that could be discussed more and put into context of additional related work.

I’m choosing my rating defensively, but I’ll be happy to raise it if the authors provide a convincing rebuttal.

---

> ### Author Response · Authors · 2022-11-14
> **Response to Reviewer qFyR**
>
> Thanks for your insightful feedback! Please see below our response to the comments.
>
> **Linear Probing Transferring to Low Data regime**
>
> > *To strengthen the paper, the authors could provide linear evaluation results or results based on an MLP head for reference and comparison with other methods, or - more sensibly - transfer results to different data sets potentially in a low data regime (see, e.g. Chen et al. 2020 for inspiration).*
>
> As discussed in the response to all reviewers, we have included in the revised Appendix Section B.3 results with linear probing and non-linear MLP heads. We have also included results on semi-supervised finetuning with $10\%$ and $1\%$ training samples following your suggestion for better assessing the model transferability.
>
>
> **Forward and Backward Passes**
>
>
> Thanks for the suggestion. We have modified Section 4.4 to explicitly show the additional forward and backward passes required by the proposed method.
>
> **Recent Work on Model Pretraining.**
>
> > *Sorting as a self-supervised task has been explored recently in [a]. I feel the paper should discuss [a] and compare their results to [a].*
>
> > *In terms of related work, it would probably be worth mentioning iGPT [b] and BigBiGAN [c] as two recent generative modeling based representation learning approaches.*
>
> Thanks for pointing out this important missing related work. We apologize that we missed this paper as it was published around the submission date.
> We have incorporated the discussion and the quantitative comparison in the revised manuscript (marked in blue).
>
> **Patch size**
>
> > *I could not find any mention of the image resolution used to train the method.*
>
> As we strive for a generic vision model pretraining framework that does not modify the target models to be trained. Therefore, we adopt the following standard settings of vision transformers: An image resolution of 224 $\times$ 224 and a patch size of $16 \times 16$.
> In the original manuscript, we mentioned the image resolution in the caption of Table 3. We have modified Appendix Section A.1 to include the image resolution as part of the implementation details (marked in blue).
> We did not find where we mentioned a patch size of 24 in our original manuscript. Please kindly let us know if there is any further confusion.
>
> **Typos**
>
> Thanks for pointing out the typos. We have fixed them in the revised manuscript.

---

> > ### Comment · Reviewer_qFyR · 2022-11-18
> > **Response to the authors**
> >
> > I thank the authors for their response. I feel the additional results strengthen the paper, so I decided to adapt the score accordingly.
> >
> > Regarding prior work on sorting, the authors still write "To the best of our knowledge, no patch-sorting based methods have been applied on ViT." in Section 3.1., so I would suggest to carefully revise the manuscript w.r.t. sorting and tone the claims down if necessary.
> >
> > Regarding gridding, the second last row of Table 1 says "Gridded (24)", and 224 is not divisible by 24. So either this is a typo, or I don't understand what "Gridded (24)" means, or there must be striding/padding involved when the authors split the input image into patches.

---

> > > ### Author Response · Authors · 2022-11-19
> > > **Response to Reviewer qFyR**
> > >
> > > Thanks for the support.
> > >
> > > **Claim**
> > >
> > > Thanks for the reminder, we have uploaded a new version of the manuscript with the claim removed.
> > >
> > > **Patch size**
> > >
> > > The masking in our framework is done by placing empty patches to the images and mask out the original content. Therefore it is not a problem if the image resolution is not divisible by the patch size, and this can be simply overcome by placing 'incomplete' patches on the edges. We also have experiments with random patches (row 3 and row 4 in Figure 3), where patches can have arbitrary aspect ratio and overlap with each other. Compared to MAE, we perform masking directly on pixels instead of dropping tokens, which places little restrictions to the patterns of the masking, and makes our method agnostic to model architectures.

---

### Official Review · Reviewer_TAyT · 2022-10-25

**Confidence:** 3
**Correctness:** 4
**Technical Novelty And Significance:** 4
**Empirical Novelty And Significance:** 1
**Recommendation:** 5

**Clarity, Quality, Novelty And Reproducibility:**

The paper was a bit difficult to read. In several places, prior knowledge about concepts involved in training energy based models was assumed from the reader. The paper proposes a novel idea of using ideas from energy based models for self supervised training.

**Strength And Weaknesses:**

Pros
	- The proposed method is simple and seems easy to implement with the provided details.
	- Strong benchmark performance improvements in the provided results on Imagenet-1K.

Cons
	- The paper was a bit difficult to read. In several places, prior knowledge about concepts involved in training energy based models was assumed from the reader. It would be helpful if more information is provided about forward and backward passes while training energy based models like the Boltzmann machines, sampling strategies, etc.
All the results were presented only on one particular dataset. Since the paper is on self-supervised pretraining of vision models, it would be really helpful if results on other vision datasets/tasks are also compared to the state-of-the-art results in the respective tasks.

**Summary Of The Paper:**

The paper proposes a novel self-supervised pretraining framework for vision models. Inspired by energy-based models (EBMs), the paper proposes training a vision model by sampling corrupted image samples and moving them along the direction of energy minimization using gradient descent, with corrupted samples having high energy and vice-versa. Comprehensive experiments were conducted using a broad range of pretext tasks with different data-corruption methods.

**Summary Of The Review:**

Borderline reject. The paper definitely proposes a novel idea, however, I think a bit more analysis and results are required. If the author(s) are able to address the above comments, the decision will be reconsidered.

---

> ### Author Response · Authors · 2022-11-14
> **Response to Reviewer TAyT**
>
> Thanks for the constructive comments! Please see below our response to the comments.
>
>
> **Background on EBMs**
>
> > *It would be helpful if more information is provided about forward and backward passes while training energy-based models like the Boltzmann machines, sampling strategies, etc.*
>
> In the original submitted manuscript, we included a brief review of implicit image generation using energy-based models with deep networks in Section 2.1, including the training (2) (3) and the sampling with backward passes (4). The background review in Section 2.1 provides the basic concepts used for inspiring our method, and we have modified the ending of Section 2.1 to suggest referring to [1, 2] for more comprehensive formulations.
>
> Note that the energy-based models in our paper refer explicitly to the line of works using energy-based models parametrized by deep neural networks for implicit modeling and sampling, directions such as Boltzmann machines do not directly fall into this category and thus are not covered in this work.
>
> **Additional Datasets**
>
> To the best of our knowledge, ImageNet is the most standard public dataset for conducting self-supervised vision model pretraining.
> In the original manuscript we submitted, we also included results on downstream model transferring with the ADE20K dataset. In the revised Appendix Section B.3, we have included new results on transferring to fine-grained datasets for better assessing the transferability of the proposed method.
>
> ---
>
> [1] Implicit generation and generalization in energy-based models. NeurIPS, 2019.
>
> [2] Improved contrastive divergence training of energy based models. ICML, 2020.

---

### Official Review · Reviewer_WmV7 · 2022-10-26

**Confidence:** 4
**Correctness:** 3
**Technical Novelty And Significance:** 2
**Empirical Novelty And Significance:** 4
**Recommendation:** 8

**Clarity, Quality, Novelty And Reproducibility:**

The writing is overall clear and the evaluations are of good quality.
Hyper-parameters are provided, but only with pseudo-code. There are some implementation details such as how the step size $\alpha$ is learned is not covered in the paper.

**Strength And Weaknesses:**

Strength
- This paper proposes a very simple baseline.
- Compared to maximum likelihood learning in EBMs, the MSE loss used here provides explicit supervision, which may reduce the instability in EBM training and potentially improve the training efficiency. Different from EBMs that are doing general generative modeling, the proposed method is doing more specific tasks on visually pretraining. Such a focus shift makes the modifications proposed to be reasonable. Though losing some nice properties of EBMs, such as flexible sampling starting from random noise, the proposed method can still serve as a strong visual pretraining model.
- From a visual pre-training perspective, combining the encoder and decoder may reduce the model complexity, yielding a more compact model with shared weights.

Weakness
- From a theoretical perspective, the contributions are limited. The overall idea is heavily based on EBMs. But this paper is doing it with a different initialization and using MSE loss as direct supervision. The MSE loss term has also been used in EBM learning in some work as a regularization term to stabilize the training.
- There is a lack of theoretical analysis. Most of the illustrations on EBMs are only intuitive, e.g., 'by gradient based updating along the direction of energy minimization, which equally encourages higher energy values for negative images, and can functionally replace the second term in (3).' From the generative modeling perspective, this loss function may or may not lead to convergence on the target distribution we are trying to model. Does the learned network have a smooth energy landscape?
- I failed to understand the experiments in Figure A. If $N=0$, how does the model obtain something different from the input? Also, the accuracy does not have large improvements even with more steps.
- In terms of efficiency, the number of epochs may not be a good metric due to the varying number of Langevin sampling steps used.
- Eq. 4 is missing the noise term. It may be better to include the original Langevin Dynamics equation here and indicate the noise term is dropped in practice.

**Summary Of The Paper:**

This paper is working on pre-training of vision models. They propose to do energy-based sampling, but with MSE loss to the original sample as supervision. Several techniques are used to improve the training efficiency, such as doing energy-based sampling starting from randomly masked images, instead of random noise. The method is evaluated on different tasks including masking, super-resolution, colorization, and sorting. Experiments on several datasets have shown better performance than some baselines.

**Summary Of The Review:**

This paper provides a simple idea with promising experimental results; It will be helpful for the community as a baseline and inspire more investigations on similar models such as diffusion models.

-------------------------------------------
I think my concerns have been addressed by the author's response. Specifically, since the generative perspective is not the major point of this paper, with some claims being toned down and adjusted a bit, I think the submission is good for acceptance and I've raised my score to 8. I think releasing the code (other than the pseudo-code alone) and adding some analysis on the generative modeling side will make the submission more helpful for the whole community.

---

> ### Author Response · Authors · 2022-11-14
> **Response to Reviewer WmV7**
>
> Thanks for the supportive comments! Please see below our response to the comments.
>
> **MSE loss term in EBM training**
>
> > *The MSE loss term has also been used in EBM learning in some work as a regularization term to stabilize the training.*
>
> We have incorporated [1] into the revised manuscript Section 3.2 (marked in blue). Please kindly suggest to us if there is additional work that uses MSE to stabilize EBM training.
>
> **Theoretical Analysis**
>
> > *There is a lack of theoretical analysis... Does the learned network have a smooth energy landscape?*
>
> We agree with the reviewer that there is no rigorous derivation from (3) to our supervision (5), therefore it may not guarantee to converge on the target distribution. However, as we have discussed in the response to all reviewers, the main focus of the paper is the discriminability and transferability instead of the generative modeling of the learned models. While we do not conduct extensive study on the smoothness of the learned energy landscape, the visualizations in the revised Appendix Section B.4 show that the learned models can distinguish real images from the faked ones, confirming the desired discriminability of the learned models.
>
>
> **Sampling Steps**
>
> > *If $N=0$, how does the model obtain something different from the input? Also, the accuracy does not have large improvements even with more steps.*
>
> In the extreme setting of $N = 0$, the experiments are simplified as binary classifications that train the model to distinguish true data samples and manually corrupted ones.
> As we have discussed in the response to all reviewers *A Generative Perspective*, we consistently observe that larger values of $N$ do not noticeably improve the image quality, meaning the condition sampling we use converges very fast at very few steps of gradient-based image recovery. Therefore, increasing the value of $N$ does not improve the performance noticeably. Note that since the generation quality of the learned models is not the main focus here, and training cost scales up linearly w.r.t. the value of $N$, we do not consider this 'early convergence' observation a problem of the proposed method.
>
> **Efficiency**
>
> > *In terms of efficiency, the number of epochs may not be a good metric due to the varying number of Langevin sampling steps used.*
>
> We agree that the number of epochs alone may not be sufficient. Therefore we included Table 5 (GPU-hours) in our original submission.
>
> **Noise term in Eq. 4**
>
> > *It may be better to include the original Langevin Dynamics equation here and indicate the noise term is dropped in practice.*
>
> In the original manuscript, we describe Eq. 4 as a gradient-based sampling instead of the original formulation of Langevin dynamics. We appreciate your comments, and we have added back the noise term in Eq. 4 and updated the corresponding paragraph to better clarify the connections.
>
>
> ---
>
> [1] Learning energy-based models by diffusion recovery likelihood. ICLR, 2021

---

> > ### Comment · Reviewer_WmV7 · 2022-11-20
> > **Some additional comments**
> >
> > Thanks for addressing my questions and concerns.
> >
> > - MSE loss term in EBM training
> >
> >     There are a few as I can recall that added the term in the released code. But I think adding one reference as a link to the EBM learning side is sufficient.
> >
> > - Generative perspective
> >
> >     I think it's fine that the major point of the paper is not on generative modeling and EBM serves as an inspiration for the design of the loss function. But drawing that connection and adding more analysis in future work may make the submission stronger.

---

### Official Review · Reviewer_74n4 · 2022-10-27

**Confidence:** 5
**Correctness:** 3
**Technical Novelty And Significance:** 3
**Empirical Novelty And Significance:** 3
**Recommendation:** 10

**Clarity, Quality, Novelty And Reproducibility:**

The idea is very easy to understand, the authors did a great job in explaining it and also I really appreciate the pytorch pseudo code available in the appendix. I don't think that there is a real novelty in this work (in the fundamental meaning) however the authors were successfully able to put the different existing pieces together which I think was something important to do.

**Strength And Weaknesses:**

Strength:
- This paper demonstrate that we can learn a model that is competitive with MAE without the need to use an explicit decoder. I really like that the authors try learn the decoder in an implicit way using score matching.
- It's also very impressive and I am really surprised that you only need 2 or 3 steps for sampling.

Weaknesses:
- **Lack of diversity in the experiments** I would have strongly appreciated the addition of the linear probing results in the paper. You mentioned lower linear probing accuracy in the limitations, but it's not clear what does that mean if you don't present any numbers. For instance [1] demonstrate that MAE can achieve 73,5% linear probing accuracy which is already better than most of SSL methods. So even if it's not as good as the performances in finetuning, it's still a competitive and respectable score (in comparison with the current literature). So does your method perform as well as MAE in linear probing (around 70%) or does it fail completely (like you only get 5% accuracy on ImageNet). Following this first point, the main issue I have with this paper is that you only use finetuning as evaluation criteria. However, anyone can also fine-tune from random weights. If your baseline in grey in table 1 and table 2, the "From Scratch" is a training from random initialized weights, it mean that the benefice of pre-training is only 2 or 3%, which is not really convincing.  So I would expect to see additional evaluation metrics like linear probing or/and KNN (even if it's not as good as MAE, it's ok but one need to be sure that the representation learned is actually "meaningful"). In addition of linear probing, you can also trained a small MLP on top of the frozen VIT backbone representation (it should be able to retrieve the information even if it's not easily linearly separable).

- **Lack of details in the experimental setup** The reconstruction in Figure 3 seems great, but are you using images from the training set or the validation set ? When you mentioned your baseline "From scratch" in Table 1 or 2, what is the training scenario of this baseline, are you also using 200 epochs or 100 epochs ? For the denoising experiments, are you still using only 2 steps ? Can you add on Figure A, the denoising experiment ? (I would expect the number of denoising steps to vary depending on which gamma parameter you are sampling from the uniform, did you try with fixed value of gamma ?) The main reason about why I am asking this, is that in the diffusion model literature one major issue is the number of sampling steps to get good image generation. When reading the paper, one could think that you were successfully able to solve the diffusion model number of sampling steps issues which seem to be difficult to believe. Are you using images of size 224x224 ? Overall, there is a lack of detail and precision in the experimental section. It's not clear why you have 7 corruption schemes in Figure 3 but you have only 3 curves in Figure A.

- **Missing important references**: Your training criterion corresponds to the denoising score matching loss [2] so this reference should at least been cited. I would have appreciated to see MSN [3] in table 3 and 5 since it's a SSL join embedding method that use masking and that is very efficient.

- **Overclaim** The authors suggest that "The proposed method can be seamlessly applied to ay deep vision models" . However, in the experimental setup, they present only results with transformers inspired vision models. What about the classic Resnet50 ? Either you specify that the proposed method can be applied on any deep transformer vision model, and present experiments with vision transformers OR you present also results with traditional architecture like Resnet/Desnet.

- **Miscellaneous** Small typo, p9 "the masked images.Different"

[1] Masked autoencoders are scalable vision learners, He et al, 2022

[2] A Connection Between Score Matching and Denoising Autoencoders, Vincent et al

[3] Masked Siamese Networks for Label-Efficient Learning, Assran et al

**Summary Of The Paper:**

This paper introduces a new series of pretext-tasks for self-supervised learning by using an energy based approach. The pretext-task is defined with a denoising score matching criterion aiming for the reconstruction of a corrupted image. The authors try several corruption schemes and show that their method perform is competitive with masked autoencoders and several other SSL methods without the need for an explicit decoder.

**Summary Of The Review:**

I really love the idea behind the paper and if the experimental setup was a bit stronger and more diverse, I would advise strongly for acceptance. However, for now, I have some concerns, especially around the potential use or not of training examples as conditioning in Figure 3 as well as for the results in linear probing which seem to have been voluntary hidden by the authors.

---

> ### Author Response · Authors · 2022-11-14
> **Response to Reviewer 74n4**
>
> Thanks for the insightful comments! Please see below our response to the comments.
>
>
> **Experiment Diversity**
>
> > *I would have strongly appreciated the addition of the linear probing results in the paper. ... In addition to linear probing, you can also train a small MLP on top of the frozen VIT backbone representation.*
>
>
> As we discussed in the response to all reviewers, and presented in the revised Appendix Section B.3, the proposed method delivers the same level of performance on linear probing as MAE. We also included a new experiment with a multi-layer non-linear head. However, we consistently find that a three-layer MLP head is insufficient to significantly improve the performance with the feature extractor frozen.
> We will keep investigating the reasons.
>
> **Experimental Details**
>
> > *... Overall, there is a lack of detail and precision in the experimental section. It's not clear why you have 7 corruption schemes in Figure 3 but you have only 3 curves in Figure A.*
>
> - We use the validation set to show the reconstructions. And the results show that the models do not exactly memorize the training set. For example, as shown in the very first example in Figure 3, instead of perfectly recovering the original image, the model 'imagines' a pot without a lid. Note that the examples here are only for the demonstration of the corresponding pre-text tasks. As we have discussed in the response to all reviewers, a powerful generative model is not the main goal of this paper.
>
> - The baseline is trained with 200 epochs of from-scratch training.
>
> - In the denoising experiments, we use: $N=2$, and the same $224 \times 224$ image resolution.
>
> - We did not include every pre-text task in Figure A and did not exhaustively try different fixed values of $\gamma$ due to the high computation cost, as they require a large amount of GPU-hours. As we have discussed in the response to all reviewers, a powerful generative model is not the goal of this paper, and scaling up $N$ cannot noticeably increase the quality of the sampled images, i.e., the results obtained in Figure 3 are nearly the best of the learned model can accomplish, and the quality does not yet meet the state-of-the-art generative models in our opinion.
> According to our recent effort after the paper submission, we tried applying the diffusion model training to EBMs and observes that sampling from scratch (a chain with initialization of $\gamma = 0$) does not work without heavily customizing the model architectures and detailed training settings. We will leave this as a direction of future efforts.
>
> **Missing References**
>
> Thanks for pointing out the important references. We have modified Section 3.2 (marked in blue) to include the discussions on [1]. And we have updated Table 3 to include the results on MSN [2].
>
> **Convolutional Architectures**
>
> In the original manuscript, we present the results on the best CNN architecture (ConvNeXt) and the best customized transformer architectures (Swin) to demonstrate that the proposed method can generalize to a wide range of model architectures. According to our early experiments, we did not find specific issues that prevented the method from generalizing to other convolutional architectures. We have included the result with ResNet in Table 6.
>
>
> **Typos**
>
> Thanks for pointing out the typos. We have fixed them in the revised manuscript.
>
> ---
>
> [1] A Connection Between Score Matching and Denoising Autoencoders, Vincent et al.
>
> [2] Masked Siamese Networks for Label-Efficient Learning, Assran et al.

---

> > ### Comment · Reviewer_74n4 · 2022-11-16
> > **Answer to Authors**
> >
> > Thanks you very much ! Really appreciate it. I am convinced now but still have some additional comment/questions (it's more about the form and way things are presented) before raising my score. Can you add the linear probe results for the Resnet50 ? (shouldn't be too long to run since you have already the pretrained resnet). Ideally, it will be good for completness of the paper to add in Table B, the evaluation results with the Resnet50/Convnext/Swin (If you can't do that before the end of the rebuttal, please do for a camera ready version).
> >
> > Concerning the hyper-parameters, you have the parameter \alpha that you initialize to 0.1 and that you learn. Can you add some detail about how this value/parameter change during training ? It is becoming very high or is it staying around 0.1 ?
> > Also when you say, you use a batch size of 256, is that 256 per gpu (so a total batch size of 2048) or is it 32 per gpu (for a total of 256 ) ?
> >
> > Also you added concerning the diffusion models:
> > > One major different is that all the inputs for the intermediate steps in our method are obtained by the previous step of restoration, instead of generated using the original images based on some noise schedulers"
> >
> > It is really not clear what you meant here and it is lacking precision. In the sampling procedure for the diffusions methods, you use as inputs the previous steps. So you start from noise, and then your model generate a given number of intermediate steps based on the previous step of restoration. The noise scheduler you are mentioning that is used to define the intermediate steps is used only for training, not for sampling which is an important distinction. Furthermore, in the original diffusion formulation in [1], all the intermediate step were used during training, so basically the model was directly trained to predict x_t+1 from x_t.  So when you say that diffusion model don't have intermediate step obtained by the previous step of restoration, it's not true if you don't specify the context.
> >
> > In addition, you are using the same noise scheduler for your input in your denoising experiment. So when you are doing your denoising experiment, with the diffusion noise scheme, you do also a gradient step from this original image based on some noise scheduler (but the main difference is that you do additional score matching step that are based on the predicted intermediate step). So basically, if N=1, your method should be equivalent to what is done in the diffusion literature when using a denoising scheduler (and without using a decoder). Do you agree ?
> >
> > So If I were you, I would mostly contrast with diffusion model by saying that you have an implicit decoder, very short chains using various type of corruptions schemes and that you add additional denoising score matching step when training the model. Also the diffusion model literature is mostly focus on generation whereas you are focusing on learning useful representation, so it might be a good place to remind that.
> >
> > You should also probably update your limitations with linear probing by adding a reference to your table in the appendix (or a foonote),
> >
> > [1] Deep Unsupervised Learning using Nonequilibrium Thermodynamics. Sohl-Dickstein et al. 2015

---

> > > ### Author Response · Authors · 2022-11-16
> > > **Response to Reviewer 74n4**
> > >
> > > Thanks for the reply to our response.
> > >
> > > **Linear probing results with ResNet-50**
> > >
> > > Directly reusing the hyperparameters we used from ViT-B, we obtain 61.7 linear probing performance with ResNet-50.
> > >
> > > We will keep studying the feature quality with different architectures and updating the discussions with more comprehensive results in the final version.
> > > We have updated the Limitation paragraph and referred to the new section in the Appendix.
> > >
> > > **Experiment details**
> > >
> > > - For the learned step size, we currently do not have the saved data to plot the curves of how the learned values of $\alpha$ change through training. By looking only at the final values after training, $\alpha$ does not change significantly from the initialized value. For example, for the model we present in the last row of Table 3, $\alpha$ converges to 0.237 from the initial value 0.1.
> > >
> > > - Following common practice, we present batch sizes as the summed batch sizes across GPUs regardless of the number of GPUs used in training.
> > >
> > > **Connection and difference to diffusion**
> > >
> > > Yes, we are in some degree talking more about training instead of sampling. And we are particularly taking about the modern diffusion models (after DDPM) where stochastic sampling is commonly used when forming the training batches. Thanks for pointing out the practice in [1] that we were previously unaware of.
> > > We agree with the reviewer that when $N=1$, i.e., no further restoration based on the outputs of the previous steps is performed in training, training in our model is similar to diffusion models. In summary, in addition to the differences pointed by the reviewer, our method has the following major practical differences compared to diffusion models (assuming a predefined non-trainable noise scheduler):
> > >
> > > - In diffusion models, a time step signal is usually fed into the model to specify the current progress of the denoising. In our method, images are the only inputs to the model.
> > >
> > > - Also related to the previous point, in diffusion models, noise scales are usually sampled from a list of values decided by the choice of noise scheduler and the number of diffusion (denoising) steps. In our denoising experiment, we sample noise scales in a truly uniform way, i.e., any value within the defined range can be sampled during training.
> > >
> > > - In diffusion models, the sampling is performed by iteratively doing *given $x_t$ predicting $x_0$* and *calculating $x_{t-1}$ based on $x_t$, predicted $x_0$, and the noise scheduler*. In our method, the sampling is performed by *repetitively refining the predictions of $x_0$*.
> > >
> > > [1] Deep Unsupervised Learning using Nonequilibrium Thermodynamics. Sohl-Dickstein et al. 2015

---

> > > > ### Comment · Reviewer_74n4 · 2022-11-16
> > > > **Answer to Authors**
> > > >
> > > > Thanks ! I am satisfied with the authors responses and adapted my score in consequence.

---

### Official Review · Reviewer_nBFT · 2022-11-02

**Confidence:** 5
**Correctness:** 3
**Technical Novelty And Significance:** 3
**Empirical Novelty And Significance:** 3
**Recommendation:** 6

**Clarity, Quality, Novelty And Reproducibility:**

Clarity: the paper is well written and easy to follow.
Quality: overall good. some statements in the paper are wrong or require further justification.
Novelty: the combination of EBMs and pretraining in the proposed way is novel. The results are interesting enough to the community.
Reproducibility: pseudo-code and implementation details are provided.

**Strength And Weaknesses:**

### Strength:

- Given the close connection between EBMs and discriminative models, it is natural to consider applying it the self-supervised learning. The proposed method enjoys the advantage of simplicity (no auxiliary networks) and efficiency (no large amount of negative samples, faster convergence).
-  Extensive experiments have been conducted to validate the effectiveness of the proposed method.

### Weakness:

- The statement that sampling from EBMs equals to argmin the energy function or Eqn. 4 is wrong. Think about a simple case where the target distribution is a 1D standard Gaussian. Then samples derived by iteratively applying Eqn. 4 will form a Dirac distribution at 0 instead of the Gaussian.
- A more principled derivation of Eqn. 5 is desired. For example, Eqn. 3 is derived by maximizing likelihood of the EBM. Does Eqn. 5 correspond to something similar but with a modified model assumption? In practice, have you ever tried to use the loss function in Eqn. 3 instead of the one in Eqn. 5, keep everything else the same and compare the results?
- The paper states that one potential reason that the method performs better than MAE is that the model is exposed to both the original images and corrupted images. It is contrary to the statement by MAE that corrupted images are away from the data manifold so it is better to let the model be exposed to only the observed part of the images. More discussion on why the statement in this paper is true is preferred.
- In the experiment of patch sorting (Table 2), it is not quite fair to train the AE baseline with position embeddings removed in pretraining and added back during fine-tuning. A fairer setting would be using randomly shuffled position embeddings in pretraining.



**Summary Of The Paper:**

This paper proposes a framework inspired by energy-based models for self-supervised pretraining. An image is first degraded by a randomly selected image corruption, and the training objective is defined as the MSE between the original image and the restored image derived by one-step gradient descent along the energy direction from the restored image from the previous sampling step. Empirical results on ViT-type structures show that the method is competitive with MAE and computationally more efficient. The proposed method can also be applied to other structures and downstream tasks.

**Summary Of The Review:**

Overall speaking, the paper proposes an interesting idea inspired by EBMs for self-supervised pretraining. The framework can handle various pretraining tasks and is agnostic to model architecture. Empirical results are promising. More principled justification of the training objective is desired.

---

> ### Author Response · Authors · 2022-11-14
> **Response to Reviewer nBFT**
>
> Thanks for the constructive feedback! Please see below our response to the comments.
>
>
> **Statement**
>
> > *The statement that sampling from EBMs equals to argmin the energy function or Eqn. 4 is wrong. Think about a simple case where the target distribution is a 1D standard Gaussian. Then samples derived by iteratively applying Eqn. 4 will form a Dirac distribution at 0 instead of the Gaussian.*
>
> Thanks for pointing this out. This sentence in the original manuscript describes a deterministic version of the sampling that ensures efficiency the most while sampling in a complex multi-modal distribution. In the original form of Langevin dynamics, a noise term helps the stochasticity of the sampling. We have modified the corresponding paragraphs and (4) for better clarification.
>
> **Derivation of Eqn. 5**
>
> > *A more principled derivation of Eqn. 5 is desired.*
>
> > *In practice, have you ever tried to use the loss function in Eqn. 3 instead of the one in Eqn. 5, keep everything else the same and compare the results?*
>
> The adopted supervision with Eqn. 5 is inspired by Eqn. 4 instead of rigorously derived from Eqn. 4. They share the same spirit of assigning lower energy scores to the faked samples. This is also the reason we term of method 'energy-inspired' instead of 'energy-based', since our method is motivated by EBMs, but does not strictly follow the practice and theory of EBMs.
>
> In the early attempt, we tried directly using Eqn. 4 as the supervision with everything else nearly identical to what we used in this paper. However, we consistently observed unstable training and the models failed to converge.
> We highlighted again that training EBMs at high resolution (224 $\times$ 224) remains an open challenge that can demand a combination of multiple techniques such as spectral normalization as in [1, 2] and customized model architectures as in [3]. Such modifications will prevent the proposed method from being a generic vision model pretraining framework. Resorting to Eqn. 5 achieves much more stable training, and the obtained energy scores exhibit desired clusters as shown in the updated Appendix Section B.4.
>
> **Statements**
>
> > *The paper states that one potential reason that the method performs better than MAE is that the model is exposed to both the original images and corrupted images. It is contrary to the statement by MAE that corrupted images are away from the data manifold so it is better to let the model be exposed to only the observed part of the images. More discussion on why the statement in this paper is true is preferred.*
>
> We are approaching self-supervised model pretraining in different ways. MAE performs mask image modeling by training the network to 'imagine' the whole images given partial observations. In our work, from an energy-score perspective, we are training the network to distinguish in-distribution and out-of-distribution images. Specifically, the intact real images serve as the in-distribution images, and the manually corrupted images (easy negatives) and the model-recovered images (hard negatives) serve as the out-of-distribution images. Therefore, it is important that the model in our framework be exposed to images from all sources for more discriminative feature learning.
>
>
> **Position Embedding in the AE baseline**
>
> > *In the experiment of patch sorting (Table 2), ..., A fairer setting would be using randomly shuffled position embeddings in pretraining.*
>
> Thanks for the suggestion. We performed a quick experiment with randomly shuffled position embedding in the AE baseline and observed that the model performs equally with the original AE baseline with the position embedding removed. We believe when shuffled position embedding is incorporated, the model learns to ignore the embedding, therefore it does not perform differently compared to the original baseline.
>
> ---
>
> [1] Implicit generation and generalization in energy-based models. NeurIPS, 2019.
>
> [2] Improved contrastive divergence training of energy based models. ICML, 2020.
>
> [3] Learning energy-based generative models via coarse-to-fine expanding and sampling. ICLR, 2020

---

### Official Review · Reviewer_A5RJ · 2022-11-04

**Confidence:** 3
**Correctness:** 3
**Technical Novelty And Significance:** 3
**Empirical Novelty And Significance:** 4
**Recommendation:** 8

**Clarity, Quality, Novelty And Reproducibility:**

Clarity:
- The paper is globally easy to understand but some parts are confusing:
- In Table 1, is "from scratch" the supervised baseline? In Table 3, what does "Mixed" refer to?

Quality:
- The introduction relies on misconceptions or small errors: it seems it refers to methods such as BYOL or SimSiam as contrastive methods whereas they are usually coined as non-contrastive (the fact that they are truly non-contrastive can be discussed but the adopted formulation is confusing). More importantly, it is exaggerated to say that the batch size is an important problem in contrastive methods: SimCLR for example can provide good results with a batch size of 1024, generally requiring 8 GPUs "only". See, e.g., [1]. Plus, non-contrastive methods such as BarlowTwins or VICReg do not require the momentum copy.

Novelty:
- To the best of my knowledge, obtaining such results on ImageNet with an EBM is new.

Reproducibility:
- No code is provided, while important hyper-parameters are given in Appendix.

[1] Garrido et al., On the duality between contrastive and non-contrastive self-supervised learning (2022).

**Strength And Weaknesses:**

Strengths:
- The proposed method seems effective: competitive results are achieved on ImageNet while using less pre-training epochs.
- As opposed to joint embeddings method, the proposed framework does not rely as much on data augmentations.
- The authors provides good results for other architectures than ViT (ConvNeXt and SwinTransformer).
- A lot of different pretext tasks are evaluated.

Weaknesses:
- Linear evaluation: the authors acknowledge that their model does not perform as well with linear probing as with fine-tuning but it seems that no numbers were provided. In my opinion it can be discussed whether linear probing is an essential feature for a SSL model, but it would be nice to have the number to better assess the proposed method.
- It would be useful to have at least one more downstream task evaluation to better assess the method.
- I understand that the outcome of the proposed method is both a discriminative and a generative model. If so, it would be useful to study the generative properties of the learned model.

Questions:
- How to ensure / do we have in practice that (5) further minimizes the energy for the final reconstruction steps than for the first steps?
- Somehow related, I understand that the $E_{\theta}$ is trained to minimize (5), meaning it is never explicitly trained to maximize the energy of out-of-domain samples?
- In the light of the last two questions, it may be interesting to provide an experimental study of the energy function to study if its behavior aligns with the motivation for EBMs.
- After the $E_{\theta} is trained, is is possible to sample "original" images from the low energy regions?
- Many SSL methods use a projector in between the pretext task (minimizing the energy) and the representation (your $\psi$), which is beneficial. Have you considered using a more sophisticated head than the linear $h$? (I do not request this experiment). Intuitively, it may improve the linear separability of the representation if this is not aligned with obtaining the energy linearly from the representation. Another quick experiment would be to try linear probing on an intermediate representation obtained before the last layer of the ViT.

**Summary Of The Paper:**

This work introduces a new self-supervised learning (SSL) framework for vision models.

The framework leverages an energy-based model (EBM): an energy $E_{\theta}$ is parameterized by a deep neural network which should learn to assign high energy to out-of-domain samples and low energy to in-domain samples. To train an EBM, one typically needs to sample from the learned distribution by doing gradient descent w.r.t $E_{\theta}$. Hence, according to the authors, training an EBM in the context of SSL is unaffordable.

To avoid this issue, the authors propose to perform conditional sampling, e.g. restoring partially corrupted data. By doing so, few gradient steps are required for sampling (2 in the paper). Then, the energy is updated by minimizing the MSE loss between the true image and successive reconstruction steps in the pixel space. The proposed framework allows to use various pretext tasks for image corruption such as masking, patch sorting, colorization, denoising and superresolution.

The authors then proceed to evaluate the proposed method by pre-training and evaluating on ImageNet1k, and finetuning on ADE20k, demonstrating on par results with other baselines while being more efficient both in terms of epochs and than most baselines. The energy is usually modeled with a ViT but ConvNeXt and SwinTransformer are also used.

**Summary Of The Review:**

This work proposes an appealing method for learning visual representations: it seems simple and provides both a generative model and a representation. Moreover, it demonstrates results on ImageNet that are on-par with many competitive methods with seemingly more efficiency in terms of epochs and FLOPs while working not only with ViTs but also with ConvNeXt. However, this comes with some flaws: for example, some parts of the paper lack clarity, the experimental analysis could study other downstream tasks, linear probing, and the generative capability of the model. I believe these problems could be adressed in the rebuttal.

________

After rebuttal: my concerns (lack of clarity on design choices, lack of discussion around the energy function), other tasks were answered and I updated my score to 8

---

> ### Author Response · Authors · 2022-11-14
> **Response to Reviewer A5RJ**
>
> Thanks for the constructive feedback. Please see below our response to the comments.
>
> **Linear Probing**
>
> > *In my opinion, it can be discussed whether linear probing is an essential feature for an SSL model, but it would be nice to have the number to better assess the proposed method.*
>
>
> We have discussed linear probing in the response to all reviewers. Our method delivers similar linear probing results compared to MAE. We have included quantitative results in the revised Appendix Section B.3.
>
> **Additional Downstream Tasks**
>
> > *It would be useful to have at least one more downstream task evaluation to better assess the method.*
>
> Thanks for the suggestion. Following the comments of reviewer **qFyR**, we have included new results in low-data regimes with semi-supervised ImageNet in the revised Appendix Section B.3.
>
> **Generative Model**
>
> > *I understand that the outcome of the proposed method is both a discriminative and a generative model. If so, it would be useful to study the generative properties of the learned model.*
>
>
> > *After the $E_\theta$ is trained, is it possible to sample "original" images from the low energy regions?*
>
> As we have discussed in *A Generative Perspective* in the response to all reviewers, while our method adopts conditional sampling for pre-text tasks of vision model pretraining, we do not design the learned model as a typical generative model.
> Based on our further studies on the train models, we believe it still requires further investigation if the learned models are capable of performing sampling unconditionally, e.g., with fully masked images or purely random noise as input conditions.
> Please note that the primary objective of this paper is a self-supervised vision model pretraining framework. Improving and evaluating the generative capacity is not the main focus here. Thus, we hope the reviewers do not penalize us for the unexplored generative capacity of the learned models.
>
>
> **Energy scores**
>
> > *How to ensure / do we have in practice that (5) further minimizes the energy for the final reconstruction steps than for the first steps?*
>
> > *Somehow related, I understand that the is trained to minimize (5), meaning it is never explicitly trained to maximize the energy of out-of-domain samples?*
>
> > *In the light of the last two questions, it may be interesting to provide an experimental study of the energy function to study if its behavior aligns with the motivation for EBMs.*
>
> In practice, as long as the sampled images at the previous step are not perfectly recovered, (5) gives the model supervision that encourages the model to assign lower energy to the real images, so that performing gradient-based adaptation to the sampled images can further push them closer to the real images. This is also ensured by the positive restriction to the learned step size $\alpha$.
>
> To further illustrate this, we have added the histogram of energy scores across steps in the Appendix Section B.4 of the revised manuscript.
> Please kindly suggest any additional experiments that can better exhibit the energy-based properties. However, the authors would like to also point out that our method is not designed to inherit all the properties of EBMs, which is the reason we term our method 'energy-inspired' instead of 'energy-based'.
>
>
> **Sophisticated Head**
>
> > *Have you considered using a more sophisticated head than the linear?*
> > *Another quick experiment would be to try linear probing on an intermediate representation obtained before the last layer of the ViT.*
>
> Thanks for the suggestions. We have added new results in Section B.3 with a multi-layer MLP as the head for a `non-linear probing' experiment. However, we consistently find that a three-layer MLP head is insufficient to significantly improve the performance with the feature extractor frozen.
> And we also observe that linear probing to the intermediate layers does not perform better than the final feature layer.
> These observations apply to both MAE and our models.
> We will keep investigating the reasons.
>
>
> **Clarifications**
>
> > *In Table 1, is "from scratch" the supervised baseline? In Table 3, what does "Mixed" refer to?*
>
> - ``From scratch'' indicates baseline with fully supervised training.
>
> - ``Mixed'' indicates models trained with a mixture of pre-text tasks discussed in this paper.
>
> Thanks for pointing these out. We have modified the corresponding caption and paragraph (marked in blue) for better clarification.
>
> > *The introduction relies on misconceptions or small errors.*
>
> Thanks for pointing out the misconceptions. We have reorganized Section 1 to better clarify the background and connections to our work. The updated content is marked in blue.

---

> > ### Comment · Reviewer_A5RJ · 2022-11-17
> > **On the MLP evaluation**
> >
> > Thank you for the clarifications.
> >
> > - I appreciate the experiments in Figure C where the energy seems to behave as expected.
> > - I also tend to agree with your remark on the generative capabilities not being the focus of the paper, although it would definitely make the submission stronger.
> > - On more experiments studying the energy landscape (not required): how would the energy generalize to other, unseen datasets of natural images? Can the energy come up with diverse and plausible restaurations of masked images when moving around the optimum (so we would only do conditional generation here)?
> > - There was a potential misunderstanding of my question on the 3-layer MLP head: I think it should also be used for training, but it seems Figure B is based on a model trained with a linear head?

---

> > > ### Author Response · Authors · 2022-11-17
> > > **Response to Reviewer A5RJ**
> > >
> > > Thanks for the support and the reply to our response.
> > >
> > > **Generalization of energy scores**
> > >
> > > We add a new set of histograms (Figure D) in the Appendix with the energy score histogram of an additional natural image dataset. The learned model assigns consistently low energy scores.
> > >
> > > **MLP head**
> > >
> > > In the early attempt, we tried using an MLP head after the feature extractor for energy score predictions, and the model did not outperform the linear-head version noticeably. As we discussed in the Introduction section, we are striving for a vision model pretraining framework with nearly no auxiliary components, we reported only the linear-head versions and did not explore exhaustively the MLP heads.
> > > We have been running another experiment with an MLP head in pretraining and the performance agrees with our early observations that it does not improve the final results noticeably.
> > >
> > > **Energy scores**
> > >
> > > > *Can the energy come up with diverse and plausible restaurations of masked images when moving around the optimum (so we would only do conditional generation here)?*
> > >
> > > Might the reviewer explain a bit more on this question so that the authors can better address the concerns? Thanks!

---

> > > > ### Comment · Reviewer_A5RJ · 2022-11-18
> > > > **Exploring the energy function**
> > > >
> > > > It was more of a question than a concern: suppose you perform reconstruction of a masked patch at inference time. To do so, you will minimize the energy function you learned and the output is Figure 3. Now, what if you move a little bit around this optimimum: would you generate other plausible reconstruction? That being said, thanks for the clarification. I updated my score accordingly.

---

> > > > > ### Author Response · Authors · 2022-11-19
> > > > > **Response to Reviewer A5RJ**
> > > > >
> > > > > Thanks for the support and further suggestions.
> > > > >
> > > > > While we are not sure about the most effective way of moving around optimums, we try manually increasing the learned step size $\alpha$ while doing conditional sampling, and find that the resulted reconstructions do not appear noticeably different. In our method, we do not encourage actively 'exploring' within the energy space, as we drop the noise term in (4) for a better sampling efficiency. We will keep exploring the learned energy landscape as a direction of future efforts.

---

### Author Response · Authors · 2022-11-14
**Reponse to All Reviewers**

We appreciate the reviewers for their support and constructive feedback on our paper. We present here our response to some questions that are asked by multiple reviewers. And we address each reviewer's concerns or questions in the corresponding response. We have uploaded the revised manuscript with updated content marked in blue.

**Linear Probing**

Our method delivers the same level of performance compared to MAE.
We did not include these results since we believe they do not demonstrate novel findings given the results reported by MAE, and the performance is lower than the contrastive learning based methods.
We have included quantitative results in the revised Appendix Section B.3.

Following the comment of reviewer **qFyR**, we further provide results in a low-data regime with semi-supervised fine-tuning in Appendix Section B.3.


**A Generative Perspective**

While our model pretraining method is inspired by EBMs, the learned model in our work does not naturally provide a strong generative model for the following reasons:

- To accelerate the training and allow for deterministic supervision with MSE, we train our models to recover images **conditionally**. It still requires further investigation if the learned models are capable of performing sampling unconditionally, e.g., with fully masked images or purely random noise as input conditions.

- Notably, we train models to recover the corrupted pixel values, and there are no steps in our methods that explicitly encourage the model to create novel contents as generative models usually do.

- We consistently observe that using larger values of $N$, i.e., performing more steps of gradient-based image recovery does not noticeably increase the quality of the reconstructed images. This is also confirmed by the results in Figure A that larger $N$ does not noticeably improve the overall performance, as later steps do not introduce many new samples. Therefore, the conditional sampling results shown in Figure 3 are nearly the best our learned models can obtain, and, in our opinion, the quality does not yet meet the generative models.

- Generating high-resolution images (224 $\times$ 224) using EBMs remains challenging, and may require customized model architectures to accomplish. However, adopting customized architectures prevents the proposed framework from being a generic vision model pretraining framework.

In summary, though we use conditional sampling for pre-text tasks of vision model pretraining under an energy-inspired framework, learning powerful generative models is not the primary objective of this work. We plan to explore its generative capability in our future work.

---

### Decision · Program_Chairs · 2023-01-20

**Decision:**

Accept: notable-top-25%

**Justification For Why Not Higher Score:**

While presenting a novel framework, this paper draws a lot from previous work. The obtained performance allows to match the other work's accuracy when finetuning the model. The linear evaluation still lags behind whole-image contrastive algorithms. For this paper to get an oral, I think the framework should unlock something that was not possible before.

**Justification For Why Not Lower Score:**

The paper is in strong shape, the experimental evaluation is thorough and the claims supported by empirical evidence.

**Metareview: Summary, Strengths And Weaknesses:**

This paper proposes a novel framework for self-supervised visual feature learning. The framework is general and encompasses ideas from previous work (masking, colorisation). The core idea is to use conditional sampling to implement an energy-based model.
The paper shows good performance, with many architectures, and using many pretext tasks.
It has been noted that the paper lacks linear evaluation, as well as additional downstream evaluations. These concerns have been partially addressed during the rebuttal.
Overall, this submission is in good shape for a publication at ICLR. It is timely, implements an interesting framework while drawing links with previous work. I recommend accepting this paper as an Spotlight.

**Note From Pc:**

if the above contains the word "oral" or "spotlight" please see: "oral" presentation means -> notable-top-5% and "spotlight" means -> notable-top-25%. As stated in our emails, we are disassociating presentation type from AC recommendations